# A novel method for quantitation of AAV genome integrity using duplex digital PCR

**Lauren Tereshko**[1][☯]*, **Xiaohui Zhao**[1][☯][¤], **Jake Gagnon**[2], **Tinchi Lin**[3], **Trevor Ewald**[1], **Yu Wang**[1], **Marina Feschenko**[1][¤], **Cullen Mason**[1]

**1** Analytical Development, Biogen, Cambridge, Massachusetts, United States of America, **2** Biostatistics, Biogen, Cambridge, Massachusetts, United States of America, **3** Analytics and Data Sciences, Biogen, Cambridge, Massachusetts, United States of America

☯ These authors contributed equally to this work.
¤ Current address: Analytical Development, Vertex Pharmaceuticals, Boston, Massachusetts, United States of America
* lauren.tereshko@biogen.com

## Abstract

Recombinant adeno-associated virus (rAAV) vectors have become a reliable strategy for delivering gene therapies. As rAAV capsid content is known to be heterogeneous, methods for rAAV characterization are critical for assessing the efficacy and safety of drug products. Multiplex digital PCR (dPCR) has emerged as a popular molecular approach for characterizing capsid content due to its high level of throughput, accuracy, and replicability. Despite growing popularity, tools to accurately analyze multiplexed data are scarce. Here, we introduce a novel statistical model to estimate genome integrity from duplex dPCR assays. This work demonstrates that use of a Poisson-multinomial mixture distribution significantly improves the accuracy and quantifiable range of duplex dPCR assays over currently available models.

## Introduction

rAAV production processes result in heterogeneous populations, where in addition to the expected encapsidated full genomes, capsids may be empty, partial, or over-packaged [1, 2]. Unwanted residual DNA from plasmids or host cells involved in the production process may also be encapsidated [1, 3, 4]. Current purification processes can efficiently separate empty from full capsids, however analytical methods that rely on separating capsid populations based on density (AUC, Cryo-EM, TEM), mass-to-charge ratios (CDMS), absorbance characteristics ($A_{260}/A_{280}$), or mass (SEC-MALS), vary in their abilities to accurately detect partially filled capsids and cannot differentiate between aberrant versus intended genomes [1–6]. It is therefore critical to develop assays to thoroughly characterize rAAV encapsidated content and genome integrity. Several next generation sequencing (NGS) platforms have been effectively used to characterize rAAV capsid content; however, use of these methods for process development is limited due to high material requirements, high cost, and the need for complex data analysis. Furthermore, NGS methods have limitations for use with non-purified samples [4, 7–11]. Single and multiplexed digital PCR (dPCR) assays have emerged as effective quantitative tools for

**Data Availability Statement:** All data files and R code are available for download from GitHub at https://github.com/tlin-biogen/genome-integrity-public.

**Funding:** This work was sponsored by Biogen. The funders had no role in study design, data collection and analysis, decision to publish, or preparation of the manuscript. All experimental design, data collection, analyses, and preparation of the manuscript were performed by Biogen employees.

**Competing interests:** The authors have declared that no competing interests exist.

characterizing encapsidated DNA given their high level of accuracy, low sample requirements and high-throughput capabilities [12–14]. dPCR assays are therefore advantageous for testing drug product and intermediate samples, and for supporting process development of early-stage gene therapy products.

dPCR technologies employ microfluidics to partition template DNA into thousands of independent amplification reactions that are expected to contain zero or one template molecule [15]. End-point fluorescence reactions result in a binary output of partitions that are either negative or positive for template. In singleplex reactions, Poisson statistics can be used to correct for the possibility of multiple templates being partitioned together and can accurately estimate the absolute concentration of template DNA [16, 17]. Reactions can be easily duplexed to assess the integrity of individual genes, or rAAV genomes by disrupting capsids prior to template partitioning, and concurrently amplifying targets at the 5' and 3' ends of the gene(s) with fluorescent probes of different wavelengths [18, 19]. With this strategy, DNA templates containing both targets are considered intact, while templates containing only one target are considered partial. Unlike singleplex reactions, Poisson statistics are not suitable to model the three-category positive data resulting from duplex reactions (5' target, 3' target, or double-positive partitions) [16].

Currently there is a divide between the current technological capability to multiplex assays and the ability to accurately analyze data from such experiments. To bridge the gap, there is a pressing unmet need to develop statistical models that can accurately quantitate end-point fluorescence data from duplex and higher-order multiplexed reactions. Here, we propose a novel Poisson-multinomial model for accurate quantitation of gene and rAAV genome integrity from duplex droplet dPCR (ddPCR) reactions using primers and probes targeting regions near the inverted terminal repeats (ITR) of the viral genomes. We compare the accuracy and precision of the model to contemporary statistical models for both plasmid and AAV samples by analyzing duplexed ddPCR data from samples across a range of genome integrities and concentrations. We demonstrate the model expands the dynamic range and improves accuracy and precision of genome integrity estimates compared to simpler models. Finally, we show that integrity values calculated with the Poisson-multinomial model have higher accuracy for both simulated and heat-fragmented rAAV material. These findings establish use of the multinomial Poisson model as a robust approach for analyzing duplex ddPCR data in support of characterizing critical quality attributes of rAAV therapies.

## Results

### Comparison of statistical models for simulated genome integrity samples using plasmid DNA

The accuracy of four analytical models was compared by using mixtures of digested viral vector plasmid (pAAV) to simulate variable degrees of genome integrity. Duplexed primer/probe sets differentially labeled with FAM and VIC dyes were designed to target the CMV enhancer (CMV) and polyadenylation (polyA) regions of the viral genome, which border the 5'- and 3'-ITRs respectively. Intact genomes were simulated by restriction enzyme digestion with MfeI, which cuts pAAV outside of the viral genome sequence. As the samples retained both target sites on a single intact template, they were assigned an expected integrity of 100%. Fragmented genomes were simulated by double-digestion of pAAV with MfeI and NheI, which bisects the viral genome sequence. As the fragmented samples contain only one target (either CMV or polyA), and zero intact templates, they were assigned an expected integrity of 0% (Fig 1 and S1 Fig). By titrating varying ratios of intact and fragmented genomes, samples were produced with either 0, 8, 18, 29, 43, 60, 82 or 100% expected integrities. The samples were then diluted

**Fig 1. Diagram of pAAV sequence.** Restriction sites are indicated by enzyme name. CMV and polyA primer/probe sets are indicated by fluorophore illustrations (blue and green respectively).

over 12 points to cover the dynamic range of the Bio-Rad QX200 ddPCR system (~1–5000 copies/μL) and duplex ddPCR was performed to assess genome integrity [17, 20]. For all experiments, pre-defined acceptance criteria were used to evaluate the suitability of the models. Relative standard deviation (RSD) of all concentrations tested for a particular sample must be <20%, and recoveries must be between 80–120% of the theoretical value.

## Calculation of pAAV genome integrity by simple percentage formula

Bio-Rad's proprietary ddPCR system outputs the number of droplets in each of four categories (double-positive, single-positive target 1, single-positive target 2, and empty droplets) and uses a Poisson distribution-based model to calculate the concentration of DNA template molecules in copies per microliter ($\lambda$), where p = the fraction of positive droplets in total droplets, and V is the average droplet volume (0.85 nL) (Formula 1, Bio-Rad, personal communication):

$$\lambda = -\ln(1 - p)/V \tag{1}$$

Variations of Formula 1 have been proposed by previous studies to calculate genome integrity as the percentage of double-positive droplets out of total positive droplets in terms of either droplet number or concentration (Formula 2) [18, 21].

$$\%\text{Integrity} = \frac{\text{Number of double positive droplets}}{\text{Number of total positive droplets}} x\, 100 \tag{2}$$

Using a Poisson distribution, it can be determined that when samples are highly dilute ($\leq$150 copies/μL), most positive droplets contain a single DNA template (probability = 0.94), and thereby most double-positive droplets will contain true intact templates as compared to the chance co-localization of 5' and 3' targets. As theorized, use of Formula 2 to calculate the genome integrity of the simulated plasmid samples was accurate over only a small portion of the theoretical dynamic range of the QX200 system. Over the tested concentration range (8–5000 copies/μL), for samples with less than 100% expected integrity, the accuracy of calculated genome integrity declined as sample concentration increased, and as expected integrity decreased (Fig 2 and Table 1).

## Calculation of pAAV genome integrity by physical linkage models

Recently, a formula for genome integrity utilizing the calculation of percent "linkage" has been suggested, where genes contained on the same template are considered linked, and genes that are physically separated are considered unlinked [22]. Linkage is defined as the number of double-positive droplets in excess of what is expected due to chance co-localization of two unlinked targets [23]. The calculation of linked target concentration is included in Bio-Rad's QuantaSoft raw data file output in terms of copies/μL.

In instances where the concentration of each target is similar, Regan et al suggest that percent linkage (i.e., genome integrity) can be calculated by dividing the linkage concentration by the average concentrations of the 5' and 3' target (CMV and polyA respectively, Formula 3) [23]. The authors note that if the concentrations of the two targets are unequal due to

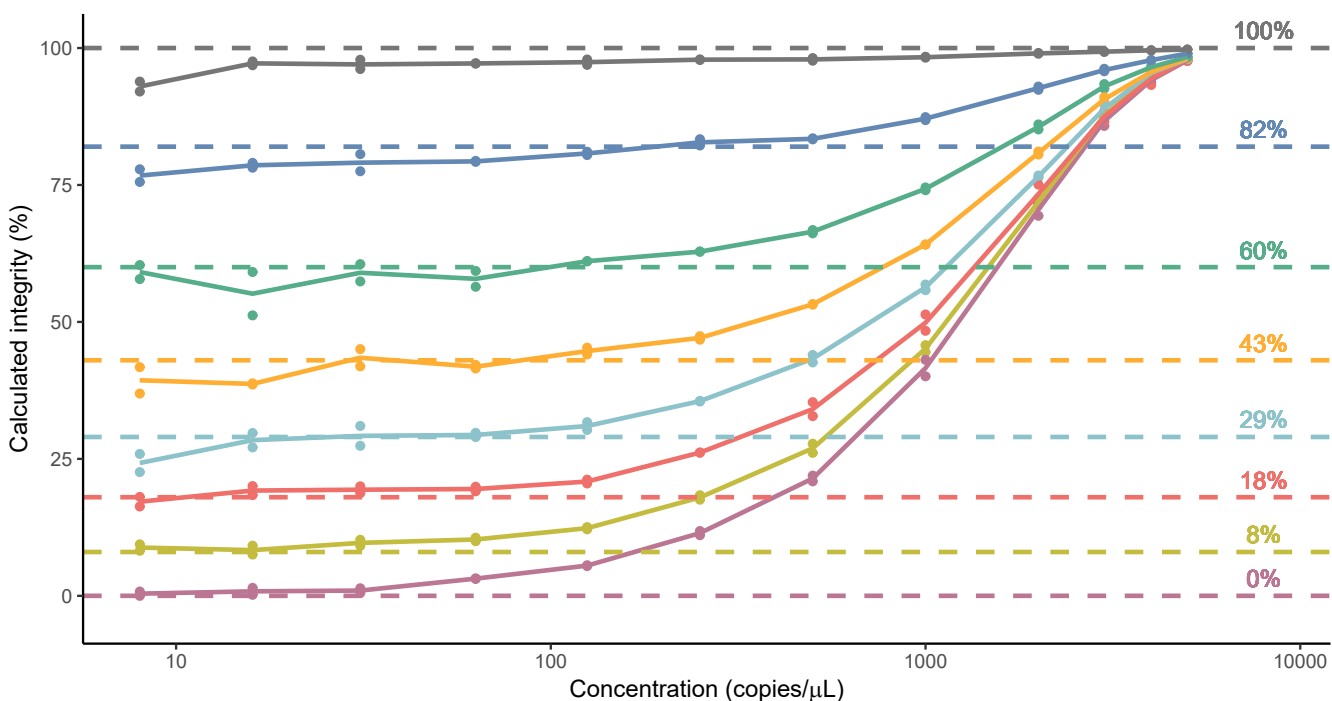

**Fig 2. Genome integrity of simulated pAAV samples calculated by Formula 2.** Genome integrity values calculated by Formula 2 are plotted as single data points (n = 1), connecting lines depict average of experimental replicates (N = 2). Dashed lines depict the expected integrity values for the samples.

amplification bias resulting from experimental conditions (method-induced genome fragmentation, differences in genome accessibility, or differences in amplicon size) that a compensated version of the equation can be used, which involves adjustment of the linkage value by addition of the absolute difference in concentrations of the 5' and 3' target, followed by division by the

**Table 1. Genome integrity and recovery of simulated pAAV samples calculated by Formula 2.**

| Copies/µL | Expected genome integrity | | | | | | | |
|---|---|---|---|---|---|---|---|---|
| | **0%** | **8%** | **18%** | **29%** | **43%** | **60%** | **82%** | **100%** |
| **8** | NC | 110.2 | 95.5 | 83.6 | 91.5 | 98.5 | 93.6 | 93.0 |
| **16** | NC | 104.3 | 106.7 | 98.0 | 90.0 | 91.9 | 95.8 | 97.2 |
| **31** | NC | 120.8 | 107.6 | 100.7 | 101.1 | 98.3 | 96.4 | 97.0 |
| **63** | NC | 128.6 | 108.3 | 101.3 | 97.3 | 96.4 | 96.7 | 97.2 |
| **125** | NC | 154.3 | 115.9 | 106.9 | 103.9 | 101.8 | 98.5 | 97.4 |
| **250** | NC | 224.5 | 145.2 | 122.5 | 109.5 | 104.7 | 101.0 | 97.9 |
| **500** | NC | 336.6 | 189.3 | 149.3 | 123.8 | 110.8 | 101.7 | 97.9 |
| **1000** | NC | 564.1 | 277.0 | 194.2 | 149.1 | 123.8 | 106.2 | 98.3 |
| **2000** | NC | 898.7 | 407.4 | 263.9 | 188.0 | 142.7 | 113.0 | 99.0 |
| **3000** | NC | 1096.6 | 486.0 | 306.7 | 210.8 | 155.0 | 117.1 | 99.3 |
| **4000** | NC | 1188.6 | 524.7 | 329.2 | 222.4 | 160.8 | 119.2 | 99.6 |
| **5000** | NC | 1224.9 | 544.3 | 337.9 | 228.1 | 164.0 | 120.7 | 99.7 |

Shaded cells indicate recoveries outside of the acceptable range (80–120%). Recoveries of 0% samples were not calculated (NC).

maximum value of either the concentration the 5' or the 3' target (Formula 4) [22].

$$\%\text{Linkage}_{\text{average}} = \frac{[\text{linkage}]}{([\text{CMV}] + [\text{polyA}])/2} \, x \, 100 \tag{3}$$

$$\%\text{Linkage}_{\text{compensated}} = \frac{([\text{linkage}] + \text{abs}([\text{CMV}] - [\text{polyA}]))}{\max([\text{CMV}] \text{ or } [\text{polyA}])} \, x \, 100 \tag{4}$$

When either Formula 3 or Formula 4 was used to calculate the percent genome integrity of the plasmid samples, the results were relatively stable across the tested concentration range (8–5000 copies/μL), as demonstrated by the assessment of the RSD ($\leq$17.1%, $\leq$27.4% respectively, Table 2). Of note, for both models, the calculated integrity values were significantly more variable as the expected integrity of the samples decreased. Additionally, the integrity values for all fragmented genome samples were overestimated as demonstrated by the calculated recoveries relative to the expected values, and the overestimations became more pronounced as expected integrity decreased (Fig 3A and 3B and Table 2). In this data set, the amplification of the 5' and 3' targets are even, and the calculated values from Formula 3 and Formula 4 are therefore expected to be similar. In contrast, Formula 3 was more accurate and precise than Formula 4 across the full range of integrity (Formula 3 accuracy 136.7%, intermediate precision 21.6%, vs Formula 4 accuracy 146.4%, intermediate precision 28.4%) (Table 2). The lack of accuracy and precision of Formula 3 for samples with lower integrity would require shortening the linear range of the method for accurate quantitation (Fig 3A, 3B and 3D).

## Calculation of pAAV genome integrity by Poisson-multinomial model

As an alternative method for genome integrity calculation, we developed a more robust statistical model that utilizes a Poisson-multinomial mixture distribution. In the model below, three categories of positive droplets are considered, with $k$ molecules in a droplet, as a multinomial model with three species containing 1.) 5' target only, 2.) 3' target only, and 3.) both 5' and 3' targets (double-positive). To model the probability of $k$ molecules within a droplet, we can utilize the binomial distribution, $p(k)$ with $k$ successes and $m$ trials, where $m$ is the total number of molecules [24]. However, when the number of droplets is large, the binomial distribution can be approximated by the Poisson distribution with parameter, lambda.

**Table 2. Genome integrity and recovery of simulated pAAV samples calculated by linkage models (Formulas 3 and 4).**

| Sample # | Expected integrity (%) | Percent Linkage$_{\text{avg}}$ (Formula 3) | | | Percent Linkage$_{\text{comp}}$ (Formula 4) | | |
|---|---|---|---|---|---|---|---|
| | | Average calculated integrity (%) | RSD (%) | Recovery (%) | Average calculated integrity (%) | RSD (%) | Recovery (%) |
| 1 | 100 | 98.3 | 0.8 | 98.3 | 99.1 | 0.5 | 99.1 |
| 2 | 82 | 88.3 | 1.0 | 107.7 | 89.3 | 1.1 | 108.8 |
| 3 | 60 | 73.4 | 2.7 | 122.3 | 74.9 | 2.6 | 124.9 |
| 4 | 43 | 58.5 | 3.3 | 136.1 | 61.4 | 5.4 | 142.8 |
| 5 | 29 | 43.8 | 6.2 | 151.1 | 46.1 | 6.1 | 159 |
| 6 | 18 | 29.0 | 14.2 | 161.1 | 30.6 | 13.5 | 169.9 |
| 7 | 8 | 14.4 | 17.1 | 180.1 | 17.6 | 27.4 | 220.6 |
| 8 | 0 | 0.8 | NC | NC | 3.9 | NC | NC |
| | | Overall accuracy | | 136.7 | | | 146.4 |
| | | Intermediate precision | | 21.6 | | | 28.4 |

Each sample was tested at twelve dilutions and the results were then averaged to produce a calculated integrity value and percent RSD. Shaded cells indicate recoveries outside of the acceptable range (80–120%) or RSD >20%. RSD and recovery not calculated for 0% expected integrity (NC).

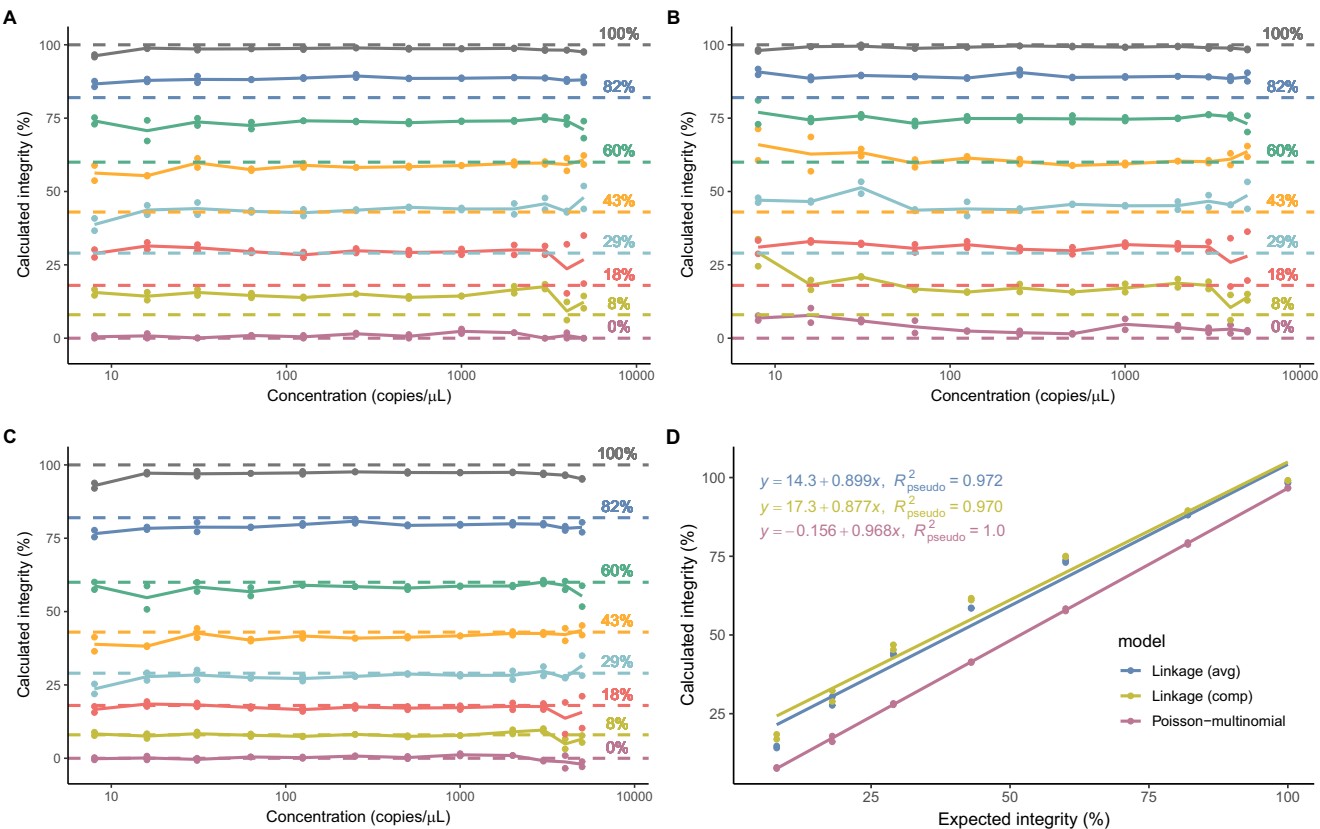

**Fig 3. Genome integrity of simulated pAAV samples calculated by linkage and Poisson-multinomial models.** A), average percent linkage (Formula 3),
B), compensated percent linkage (Formula 4). C), Poisson-multinomial (Formula 7). Calculated percent integrity values are plotted as single data points
(n = 1), connecting lines depict average of experimental replicates (N = 2). Dashed lines depict the expected values for the samples. D), Linearity of models.
Average calculated percent integrity values from sample dilutions are plotted as points (n = 12, N = 2).

This Poisson-multinomial model has two unknown parameters, $p_{CMV}$ and $p_{polyA}$. To estimate these parameters, the measured numbers of CMV single-positive droplets, polyA single-positive droplets, and the total number of droplets are needed. The upper limit of the dynamic range of the QX200 ddPCR plate reader is 5000 copies/μL. Given that the average droplet volume is 0.85 nL, this upper limit is equal to 4.25 copies/droplet. It can be further determined by the Poisson distribution that the probability of more than 20 copies of template molecules being present in a droplet is negligible (probability = 5.4E−9). It is therefore numerically sufficient to model the situation where the average number of template molecules within a droplet is ≤ 20 copies.

The calculated number of CMV single positive-droplets across all droplets can be derived from the Poisson-multinomial model where $k$ = the number of template molecules within a droplet, and $D$ is the total number of droplets (Formula 5).

$$\left(\sum_{k=1}^{20} p(k) \times p_{CMV}^{k}\right) \times D = \text{CMV single positive droplets} \tag{5}$$

Similarly, the calculated number of polyA single-positive droplets across all droplets can be derived as Formula 6.

$$\left(\sum_{k=1}^{20} p(k) \times p_{polyA}^{k}\right) \times D = \text{polyA single positive droplets} \tag{6}$$

Solving the 20th degree polynomial gives us the unknown quantities $p_{CMV}$ and $p_{polyA}$. Genome integrity can then be calculated as the expected (E) number of full genomes across all droplets divided by the expected number of total DNA template molecules across total droplets:

$$E[\text{Full genome}]/E[\text{Total DNA template}]$$

where the numerator quantity is derived from the double expectation rule:

$$E[E[\text{Full genome given k molecules in a droplet}]] = (1 - p_{CMV} - p_{polyA}) \times \lambda \times D$$

and the denominator is given by:

$$\lambda \times D$$

With these expressions, Formula 7 can be used to calculate the percentage of templates that are fully intact:

$$\%\text{Integrity} = (1 - p_{CMV} - p_{polyA}) \times 100\% \tag{7}$$

When the Poisson-multinomial model (Formula 7) was used to calculate the percent genome integrity of the plasmid mock samples, all values were consistent (RSD $\leq$18%) and aligned with the expected values (recoveries between 94.3–96.9%) across the tested concentration range (8–5000 copies/µL) (Fig 3C and Table 3). The overall accuracy of the Poisson-multinomial distribution model was 96.3% and the accuracy was consistent across the full range of intact genomes (0–100%). Although the calculated integrity values were significantly more variable as the integrity of the simulated plasmid samples decreased, the Poisson-multinomial model was more precise than linkage models (Poisson-multinomial intermediate precision <1%, vs >21% and 28% for linkage models), as calculated by the RSD of all the recoveries (Tables 2 and 3). Plotting the sample integrity versus the calculated integrity values for each model shows that the Poisson-multinomial model is more linear than linkage models in the range of 8–100% when the data were fit using generalized least squares (GLS) to account for correlations among experimental replicates (Fig 3D, S2 Fig and S1 Table).

**Table 3. Genome integrity and recovery of simulated pAAV samples calculated by the Poisson-multinomial distribution model.**

| Sample # | Expected Integrity (%) | Average Calculated integrity (%) | RSD (%) | Recovery (%) |
|---|---|---|---|---|
| 1 | 100 | 96.7 | 1.4 | 96.7 |
| 2 | 82 | 79.1 | 1.7 | 96.4 |
| 3 | 60 | 58.0 | 4.1 | 96.6 |
| 4 | 43 | 41.4 | 4.8 | 96.2 |
| 5 | 29 | 28.0 | 8.2 | 96.7 |
| 6 | 18 | 17.0 | 15.8 | 94.3 |
| 7 | 8 | 7.8 | 18.0 | 96.9 |
| 8 | 0 | -0.1 | NC | NC |
| | | | Overall accuracy | 96.3 |
| | | | Intermediate precision | 0.9% |

Each sample was tested at twelve dilutions and the results were then averaged to produce a calculated integrity value and percent RSD. There were no recoveries outside of the acceptable range (80–120%) or RSD >20%. RSD. Percent recovery and RSD not calculated for 0% expected integrity (NC).

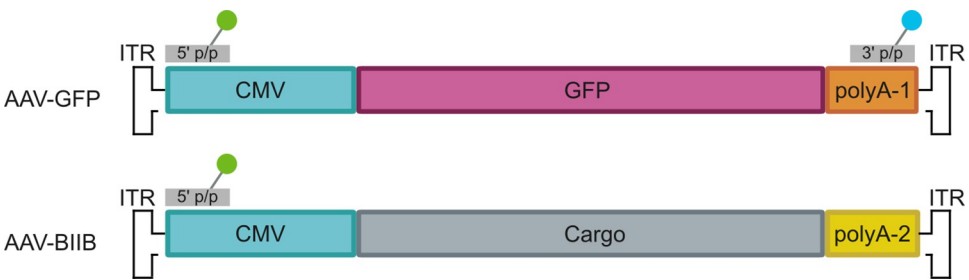

**Fig 4. Diagram of rAAV sequences.** AAV-GFP and AAV-BIIB genomes share a common promoter (CMV) but different polyA sequences. 5' (green) and 3' (blue) primer/probe sets are indicated by fluorophore illustrations.

## Comparison of linkage vs Poisson-multinomial models for simulated integrity samples using rAAV material

To evaluate the Poisson-multinomial model for genome integrity empirically with rAAV material, fragmented rAAV genomes were simulated by combining two samples (AAV-GFP and AAV-BIIB) containing common promoter sequences (CMV) but differing polyA sequences. Primer/probe sets were designed to target the shared enhancer regions of both samples, and the polyA sequence of AAV-GFP (polyA-1, Fig 4). Since AAV-GFP is a heterogenous mixture containing some partial genomes, it was first assayed with both primer/probe sets in a duplex reaction to estimate genome integrity from dilutions <250 copies/μL using Formula 2. The calculated integrity of AAV-GFP (84%) was then set as the maximal expected integrity value. When assayed with the same primer/probe sets, AAV-BIIB is expected to have no amplification with the polyA-1 targeting primer/probe and is considered 0% intact. In this manner, six mock integrity samples were prepared by mixing AAV-GFP and AAV-BIIB in varying concentration ratios to produce samples with either 0, 5, 10, 21, 42, 63 or 84% of genomes expected to contain both ddPCR targets.

It is possible that rAAV samples may contain both targets within a single capsid without the targets being physically linked. To truly assess genome integrity as opposed to double-positive capsids, the samples must be decapsidated prior to droplet generation. Decapsidation was performed by alkaline lysis and a short incubation at low heat (10 minutes, 60°C) to minimize hydrolysis of the phosphate DNA backbone [22, 25–31]. Duplexed ddPCR assays were performed on the rAAV samples as described for pAAV samples, but tested over four dilutions that spanned a narrower range of concentrations (62–498 copies/μL).

The genome integrity of each sample was calculated using either average linkage percentage (Formula 3) or the Poisson-multinomial model (Formula 7). Comparison with the compensated linkage percentage (Formula 4) was not included as the concentrations of CMV and polyA-1 are expected to be uneven due to the experimental design. Both Formula 3 and Formula 7 yielded consistent results across the range of concentrations tested, as evidenced by the flat lines for each simulated integrity sample (Fig 5A and 5B). The integrity values calculated by Formula 3, however, were consistently overestimated compared to those produced by the Poisson-multinomial model. (Fig 5A and 5B).

To compare the accuracy and precision of the models, the percent recoveries of calculated genome integrities were evaluated in six independent experiments. When integrity was calculated using the average percent linkage model, all sample replicates showed acceptable precision (RSD ≤8.5%), however the majority of samples below 84% expected integrity over-recovered (>120%) (Table 4). Consistent with what was observed with the simulated plasmid samples, over-estimation of integrity worsened as the expected integrity of the simulated

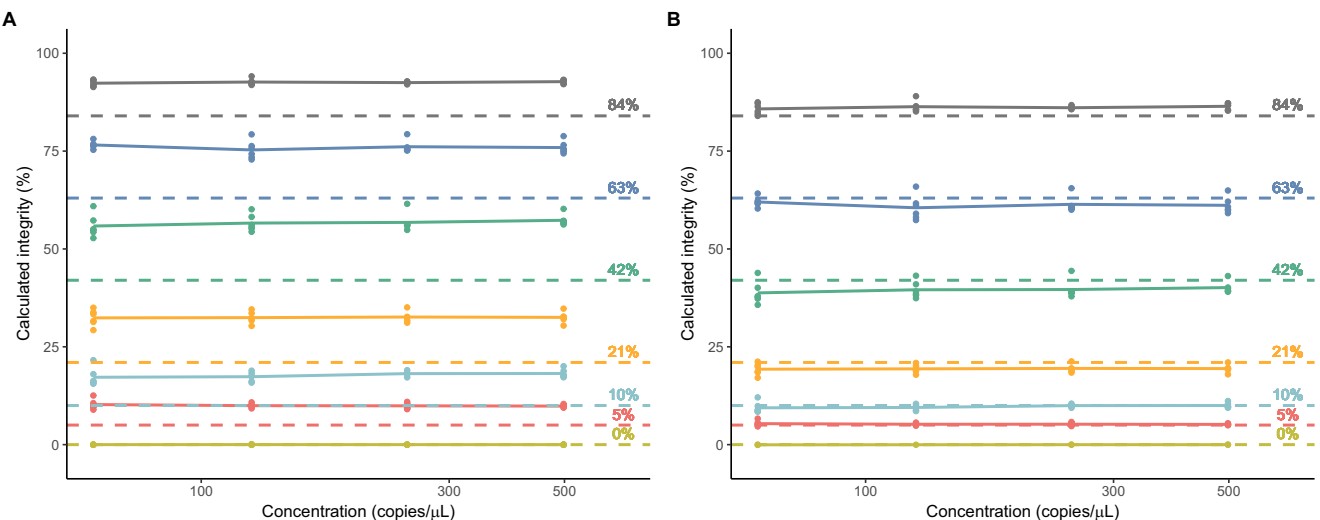

**Fig 5. Comparison of calculated genome integrities of simulated rAAV samples.** Percent integrity was calculated with A.) average percent linkage model (Formula 3) or B.) Poisson-multinomial model (Formula 7). Calculated percent integrity values are plotted as single data points (n = 1), connecting lines depict average of experimental replicates (N = 6). Dashed lines depict the expected values for the samples.

rAAV samples decreased (Tables 2 and 4). In comparison, when the Poisson-multinomial model was used, all sample replicates demonstrated acceptable recovery (86.6–117.8%) and precision (RSD<9%), as summarized in Table 4., The Poisson-multinomial model showed better overall accuracy and intermediate precision when compared with the average percent linkage model (accuracy: 98.1% vs 149.6%, intermediate precision: 6.5% vs 21.8% respectively).

To compare the linearity of the models, the average calculated percent genome integrity of each sample replicate (summarized in Table 4) was plotted against the expected integrity value, and the data were fit using GLS to account for correlations among experimental replicates. Although both models showed linear sample recovery in the range of 5–84%, the Poisson-multinomial data had a better linear fit with a slope of 1.01 and pseudo $R^2$ of 0.996 (Fig 6, S3 Fig and S2 Table).

## Comparison of linkage vs Poisson-multinomial models for genome integrity of heat-fragmented rAAV material

To evaluate the Poisson-multinomial model in a more realistic context, we wanted to generate genomes of varying integrity from a single source of rAAV material. Thermal stress has been shown to degrade DNA through spontaneous hydrolysis [22, 25–31]. Replicate AAV-BIIB samples were therefore decapsidated by alkaline lysis and followed by either a 0, 1, 5, 10, 20, or 30-minute incubation at 95˚C to generate variably intact, heat-fragmentated genomes. Primer/probe sets targeting the CMV enhancer and polyA-2 sequences were used to assay the samples in duplex reactions. Sample material was previously determined to have an estimated integrity of 48% via single-molecule long-read NGS.

Compared to the NGS data, both linkage-based models (Formula 3 or Formula 4) had greater overestimated integrity of the unheated sample material, than the Poisson-multinomial model (Fig 7). Additionally, although genome integrity decreased as incubation time at 95˚C increased for all models, the values were much higher when calculated by either linkage-based model compared to the Poisson-multinomial model. Of note, the raw concentration of each

**Table 4. Comparison of average genome integrities and percent recoveries calculated via Formula 3 and the Poisson-multinomial model for simulated rAAV samples.**

| Sample # | Expected Integrity (%) | Percent Linkage$_{avg}$ | | | Poisson-multinomial | | |
|---|---|---|---|---|---|---|---|
| | | Average replicate integrity (%) | RSD (%) | Recovery (%) | Average replicate integrity (%) | RSD (%) | Recovery (%) |
| 1 | 84 | 93.4 | 0.6 | 111.1 | 87.6 | 1.1 | 104.3 |
| | | 92.3 | 0.8 | 109.9 | 85.5 | 1.3 | 101.8 |
| | | 92.2 | 0.6 | 109.7 | 85.6 | 1.1 | 101.9 |
| | | 92.5 | 0.5 | 110.1 | 86.2 | 0.9 | 102.7 |
| | | 92.7 | 0.4 | 110.3 | 86.3 | 0.8 | 102.7 |
| | | 92.4 | 0.4 | 110.0 | 85.8 | 0.8 | 102.1 |
| 2 | 63 | 78.9 | 0.7 | 125.2 | 65.1 | 1.2 | 103.4 |
| | | 76.3 | 0.3 | 121.1 | 61.6 | 0.7 | 97.8 |
| | | 75.0 | 1.4 | 119.0 | 60.0 | 2.2 | 95.3 |
| | | 75.0 | 2.2 | 119.0 | 60.0 | 3.4 | 95.2 |
| | | 75.2 | 1.1 | 119.4 | 60.2 | 1.8 | 95.6 |
| | | 75.5 | 1.1 | 119.9 | 60.7 | 1.9 | 96.4 |
| 3 | 42 | 60.7 | 1.1 | 144.6 | 43.6 | 1.4 | 103.9 |
| | | 56.0 | 4.2 | 133.4 | 38.9 | 5.8 | 92.5 |
| | | 55.6 | 1.7 | 132.4 | 38.5 | 2.2 | 91.6 |
| | | 55.5 | 2.0 | 132.2 | 38.4 | 2.5 | 91.4 |
| | | 56.2 | 1.5 | 133.8 | 39.0 | 2.1 | 92.9 |
| | | 55.8 | 1.9 | 132.9 | 38.8 | 2.8 | 92.3 |
| 4 | 21 | 34.8 | 0.7 | 165.9 | 21.1 | 0.8 | 100.5 |
| | | 31.6 | 5.1 | 150.4 | 18.7 | 6.0 | 89.2 |
| | | 30.8 | 1.7 | 146.7 | 18.2 | 1.8 | 86.6 |
| | | 32.0 | 1.4 | 152.6 | 19.1 | 2.0 | 90.9 |
| | | 32.7 | 1.5 | 155.6 | 19.5 | 1.6 | 92.9 |
| | | 33.0 | 2.6 | 157.1 | 19.7 | 2.8 | 94.0 |
| 5 | 10 | 19.7 | 7.3 | 197.3 | 10.9 | 8.1 | 109.4 |
| | | 18.1 | 6.8 | 180.7 | 9.9 | 7.4 | 99.4 |
| | | 17.2 | 6.4 | 172.0 | 9.4 | 7.1 | 94.1 |
| | | 17.0 | 5.3 | 170.1 | 9.3 | 5.8 | 92.8 |
| | | 17.5 | 5.4 | 174.6 | 9.6 | 5.6 | 95.7 |
| | | 16.9 | 6.4 | 168.6 | 9.2 | 7.2 | 91.9 |
| 6 | 5 | 11.2 | 8.5 | 223.6 | 5.9 | 9.0 | 117.8 |
| | | 10.0 | 3.8 | 200.5 | 5.3 | 4.0 | 105.5 |
| | | 9.7 | 3.9 | 194.9 | 5.1 | 3.5 | 102.0 |
| | | 9.6 | 6.7 | 192.7 | 5.1 | 7.4 | 101.1 |
| | | 9.8 | 6.5 | 196.8 | 5.2 | 6.7 | 103.4 |
| | | 9.5 | 1.4 | 190.3 | 5.0 | 1.3 | 99.5 |
| 7 | 0 | 0.0 | NC | NC | NC | NC | NC |
| | | 0.0 | NC | NC | NC | NC | NC |
| | | 0.0 | NC | NC | NC | NC | NC |
| | | 0.0 | NC | NC | NC | NC | NC |
| | | 0.0 | NC | NC | NC | NC | NC |
| | | 0.0 | NC | NC | NC | NC | NC |
| | | | | **Overall accuracy** | **149.6** | | **98.1** |
| | | | | **Intermediate precision** | **21.8** | | **6.5** |

Highlighted cells are outside the acceptable recovery range (80–120%). RSD and recovery not calculated for 0% expected integrity (NC).

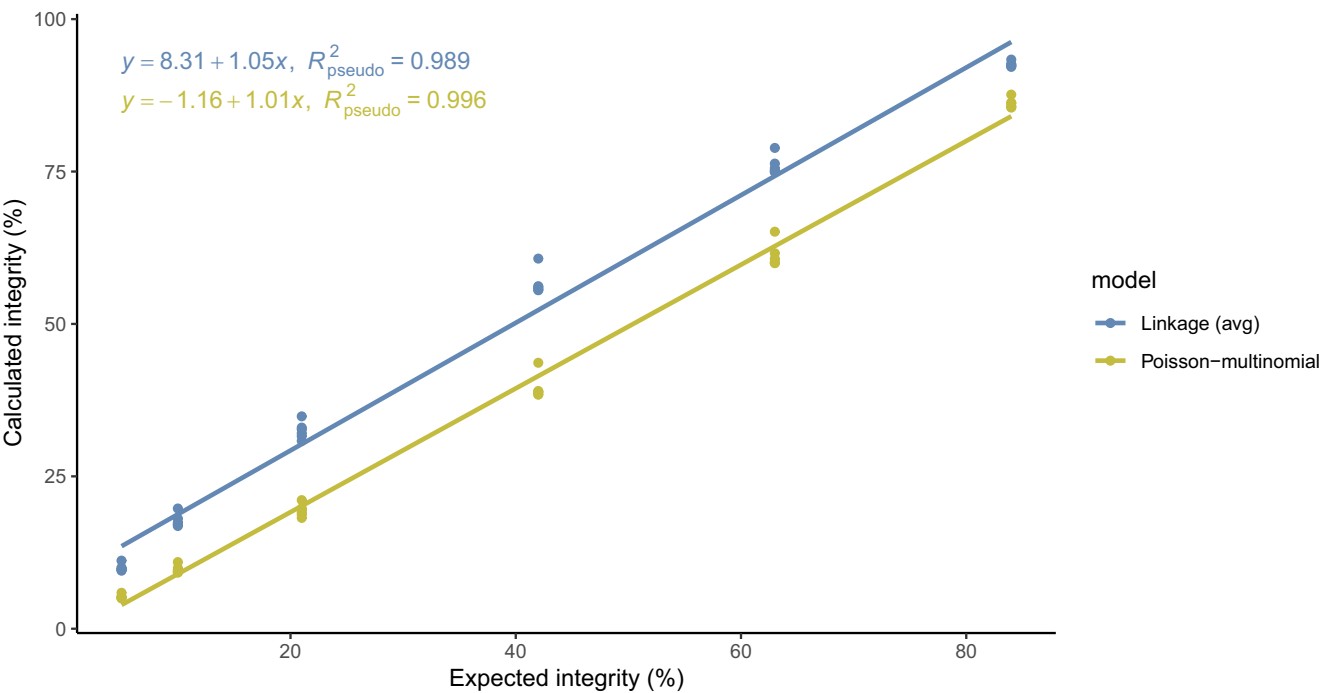

**Fig 6. Linearity of the average percent linkage model (Formula 3) and Poisson-multinomial model for rAAV simulated genome integrity samples.**
Average calculated percent integrity values from experimental replicates are plotted as points (n = 4, N = 6).

single ddPCR target remained consistent across incubation time, demonstrating that heat treatment of up to 30 minutes leads to increased fragmentation, and not complete degradation of the template or reannealing of the ssDNA.

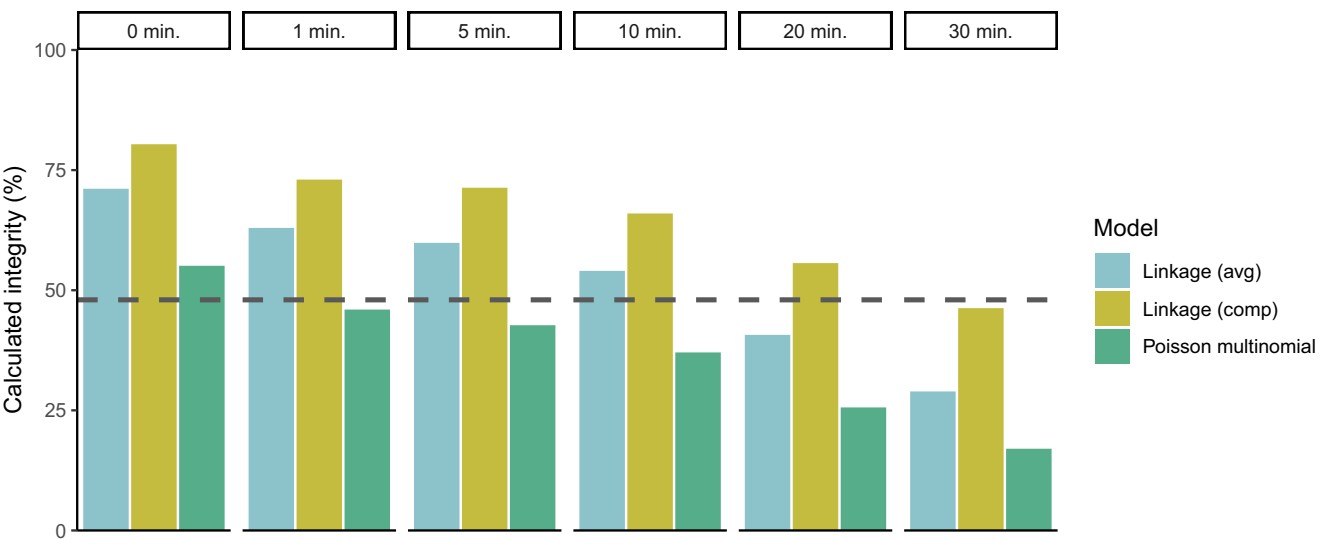

**Fig 7. Genome integrities of heat-fragmented rAAV calculated by linkage or Poisson-multinomial models.** Average calculated genome integrity values are plotted as single bars (n = 12, N = 2). Dashed line depicts expected integrity as determined by long-read NGS. Average percent linkage (Formula 3), compensated percent linkage (Formula 4).

## Discussion

This work summarizes the development of a novel analytical model for the calculation of gene and genome integrity from duplexed dPCR assays. The new model has been coded in R as a Shiny application, and the code is publicly available. Use of the Poisson-multinomial mixture distribution offers more rigorous modeling compared to simplistic percentage-based calculations (described by Formula 2), which are accurate only at highly dilute concentrations. Comparatively, our model is highly accurate across a wide range of template concentrations, expanding the dynamic concentration range of duplex integrity assays to at least four orders.

Unlike Poisson models which are limited to the prediction of single categories of droplet species (positive or negative), the Poisson-multinomial model considers the probability of multiple species of positive droplets, making it more suitable for the interpretation of duplexed data. Although linkage models also consider multiple species of positive droplets by subtracting the number of droplets with linked targets expected by chance from the number of double-positive droplets, we found our model to have improved intermediate precision and accuracy (compared to Formulas 3 and 4). Critically, use of linkage models consistently resulted in inflated integrity values for both simulated integrity samples (plasmid and rAAV) and heat-degraded rAAV samples.

We speculate linkage models may over-estimate genome integrity by calculating the fraction of full genomes (concentration of linked targets) out of the concentration of genome fragments (as either the average or maximum of 5' and 3' target concentrations) when the total concentration of both targets (sum of 5' and 3' target concentrations, minus linked target concentration) should be considered instead. The linkage models may have been developed with the assumption that even if genomes are fragmented that both 5' and 3' regions will be present, when in fact, truncated genomes often have fragmentation bias. Capsids containing truncated rAAV genomes can result from a variety of reasons such as defective replication or errors in viral packaging [22–24]. Encapsidation errors have been shown to have bias towards 5' truncation [24, 25]. Furthermore, the type and rate of truncation event is also dependent on the production method, vector sequence, and whether the virus is single-stranded or self-complementary [26–28]. For these reasons, we believe it is more appropriate to consider the total concentration of template DNA as opposed to the average (or maximum) of 5' and 3' targets.

Accurately characterizing the encapsidated DNA of gene therapy material is a critical aspect of drug product safety and efficacy. Duplex dPCR assays have the potential to increase throughput and lower the cost of molecular characterization relative to NGS-based methods. By targeting regions near each ITR, duplex dPCR assays can provide accurate estimates of genome integrity and offer the advantage of absolute quantitation without the sample preparation bias that is inherent to NGS [32–34]. Despite advancements in single-molecule long-read technologies such as PacBio and Nanopore, these platforms also exhibit some degree of sequencing biases that can impact quantitative results [4, 10, 34, 35]. Additionally, duplex dPCR reactions require minimal material for testing, can accommodate less purified samples and are relatively high throughput [12–14]. Together, high levels of accuracy, precision, and sensitivity, combined with high throughput, makes duplex dPCR assays useful methods for characterizing rAAV and as an orthogonal approach to NGS. By expanding both the concentration dynamic range, and the linear range of the duplex dPCR integrity values to cover >0 to 100%, the Poisson-multinomial model is well-suited to serve as a screening method to support DOE studies and guide process decisions. Despite these strengths, it is important to note that duplex dPCR assays detect only the presence or absence of the primer and probe targets, and do not provide information about empty capsids, or encapsidated DNA sequences. This

limitation underscores the need for orthogonal methods such as EM, CDMS, AUC, SEC-MALS, and NGS to fully characterize capsid populations, rAAV genomes, and contaminants.

The work summarized here for quantitating genome integrity can be applied to other duplex dPCR assays, including those aimed at monitoring residual DNA size and identity. FDA guidance recommends that residual DNAs be limited to under 10 ng/dose and less than 200 base pairs in length in final drug product [36]. As contaminants are expected to be present in very low concentrations in final drug products, utilizing the Poisson-multinomial model for accurate analysis of residual DNA integrity may be a useful tool in assessing the integrity and therefore risk level of contaminant DNAs. Multiplexed dPCR methods could potentially be expanded beyond duplex reactions to simultaneously measure both the quantity and integrity of specific residual DNAs from plasmids or host cell genes. By controlling whether decapsidation occurs prior to, or after droplet generation, one could potentially determine the levels of encapsidated residuals relative to the rAAV genome and provide characterization data around DNA content in different capsid populations: those that contain the rAAV genome and those that do not. Such analyses would require both the viral genome and residual plasmids to be detectable at similar dilutions, for which the expanded dynamic range of the Poisson-multinomial model may be advantageous. Currently, limited data is available comparing quantification of contaminant DNA using dPCR vs NGS. We look forward to future progress showing the correlation of orthogonal methods and the continued advancement of molecular technologies.

## Materials and methods

### Plasmid and rAAV material

pAAV material was produced by Biogen. AAV-GFP was purchased from Charles River Laboratories (CV10006). AAV-BIIB material was produced by Biogen using a transient transfection process.

### Primers and probes

Custom primer and TaqMan probe sets targeting cytomegalovirus enhancer (CMV), and poly-adenylation signals were purchased from Thermo Fisher Scientific. Sequences targeting CMV were designed as Forward Primer: 5′-ATGGAGTTCCGCGTTACAT-3′; Reverse Primer: 5′-AGTCCCTATTGGCGTTACTATG-3′; Probe: 5′-FAM- AACTTACGGTAAATGGCCCGCCT-MGB/NFQ-3′ (pAAV and heat fragmentation experiments) and 5′-VIC-AACTTACGGTAAATGGCCCGCCT-MGB/NFQ-3′ (simulated rAAV experiment). Sequences targeting pAAV polyA were designed as Forward Primer: 5′-GTTGGGAAGACAACCTGTAGGG-3′; Reverse Primer: 5′-GATTGCAGTGAGCCAAGATTG-3′; Probe: 5′-VIC-TCCAGCTTGGTTCCCAATAGACCC-MGB/NFQ-3′. Sequences targeting the AAV-GFP polyA (polyA-1 in Fig 4) were designed as Forward Primer: 5′-AGCAATAGCATCACAAATTTCACAA-3′; Reverse Primer: 5′-CCGCGTTAAGATACATTGATGAGTT-3′; Probe: 5′-FAM-AGCATTTTTTTCACTGCATTCTAGTTGTGGTTTGTC-MGB/NFQ-3′. Sequences targeting AAV-BIIB polyA (polyA-2 in Fig 4) were designed as Forward Primer: 5′-GCCAGCCATCTGTTGTTTGC-3′; Reverse Primer: 5′-GCGATGCAATTTCCTCATTT-3′; Probe: 5′-VIC-TTAGGAAAGGACAGTGGGAGTGGC-MGB/NFQ-3′.

### Sample preparation

**Plasmid simulated genomes.** pAAV was digested with MfeI alone or with MfeI and NheI together (New England Biolabs). DNA concentrations of digests were measured using a Nano-drop spectrophotometer (Thermo Fisher Scientific) and converted to copies/mL. Digested

plasmids (MfeI and MfeI/NheI) were diluted in TE and mixed at varying ratios to simulate varying degrees of genome integrity.

**rAAV simulated genomes.** Simulated integrity samples were prepared by mixing varying ratios of two rAAVs (AAV-GFP and AAV-BIIB) containing the same promoter (CMV) and different polyA tails (polyA-1 and polyA-2 respectively). Samples were pre-diluted to a target concentration range (1.125e5-5e6 vg/μL) and digested with DNase I for 30 minutes at 37˚C. Viral capsids were disrupted with SDS solution and incubation at low heat (10 minutes, 60˚C).

**Heat fragmented rAAV genomes.** AAV-BIIB samples were pre-diluted, DNase treated and decapsidated following the procedure described for rAAV simulated genomes. Following decapsidation, samples were incubated at 95˚C for either 0, 1, 5, 10, 20 or 30 minutes.

## ddPCR

Following sample preparation, samples were serially diluted and combined with ddPCR master mix with the addition of SmaI (New England Biolabs). Samples were partitioned into approximately 20,000 droplets using a Bio-Rad Automated Droplet Generator (1864101). Droplets were subjected to ITR restriction digestion and endpoint PCR thermal cycling in a BioRad C1000 Touch thermocycler programmed to follow: 1 cycle of 37˚C × 15'; 1 cycle of 95˚C × 10'; 40 cycles of 94˚C × 30", 60˚C × 1', and 1 cycle of: 98˚C × 10'; 4˚C hold). Samples were read on a Bio-Rad QX200 droplet reader.

## Data analysis

Droplet analysis was performed using BioRad QuantaSoft software (version 1.7.4). Inter-assay replicates and experimental replicates are indicated in the figure legends. Percent recovery was calculated by comparing the calculated genome integrity to the expected genome integrity using the following equation: $\%\text{Recovery} = \frac{\text{calculated integrity}}{\text{expected integrity}} x\ 100$

Percent relative standard deviation (RSD) was calculated as the standard deviation of the calculated integrity replicates divided by the average of the calculated integrity replicates, multiplied by 100. Assay accuracy was calculated as the grand average of the replicate average percent recovery values. Intermediate precision was calculated as the RSD of the replicate average percent recovery values. Pre-defined acceptance criteria of RSD <20% and recoveries between 80–120% were used for data analysis of all experiments.

## Long-read NGS

1-2E+12 vg of AAV-BIIB material was DNase treated and lysed with SDS solution. Extracted ssDNA was annealed by rapid heating and cooling using a C1000 Touch thermocycler (BioRad) to form dsDNA and purified using a QIAquick PCR purification kit (Qiagen). Independent libraries of rAAV were prepared as 1 μg samples diluted to 50 ng/μL with 20 ng of λDNA-BstEII digest (New England Biolabs) added for fragment size controls [4] and sequenced on two instruments: MinION Mk1B (Oxford Nanopore Technologies) and Sequel IIe (PacBio) following manufacturers guidelines for Native Barcoding and Ligation sequencing kits and SMRTbell prep kits respectively. A custom python-based analytical pipeline was used to pre-process Nanopore reads as FASTQ files. All sequencing reads were aligned with BWA-MEM [37], low-quality alignments were removed with SAMtools [38]. Percent genome integrity was estimated using the analytical pipeline that defined full-length reads as >90% ITR-ITR coverage, and calculated integrity as: $\%\text{Integrity} = \frac{\text{number full length reads}}{\text{number of positive reads}} x\ 100$. Genome integrity results from both sequencers (46% Nanopore, 49% PacBio) were averaged and used as the expected integrity (48%) for AAV-BIIB genome in heat fragmentation experiments.

### Shiny app for Poisson multinomial model

A Web-based Shiny application was developed and deployed on the cloud-based Posit platform [39, 40]. The code for the app and raw data can be accessed publicly at: https://github.com/tlin-biogen/genome-integrity-public

### Shiny app description

Data files generated by the QX200 droplet reader (BioRad) are imported into the app which extracts relevant raw data to calculate genome integrity with the Poisson multinomial model. Each well in the assay plate has two rows which correspond to two respective targets. Columns B through L are raw data extracted from the raw data file, while genome integrity is appended in column M in duplicate rows for each well. It is worth noting that "NA" will be displayed in column M when any droplet number in columns I-K is 0, as it fails to meet the prerequisites for Poisson-multinomial model. To ensure data integrity, the application verifies that the two percent full genome values in duplicate rows are identical, which is displayed as Boolean values in column N.

## Supporting information

**S1 Fig. Confirmation of pAAV restriction digest.** 1.5 μg of pAAV was digested with either MfeI alone, or with MfeI and NheI together for 1 hour at 37˚C. Samples were run on a 1% agarose gel and visualized with SYBR Safe (Invitrogen) on an Odessey M imaging system (Licor). From left to right: Lane 1: Thermo 1KB ladder (10787018); Lane 2: undigested pAAV; Lane 3: pAAV MfeI digest; Lane 4: pAAV Mfe, NheI double-digest. Digested bands were of expected sizes (Lane 3: 7284, 4266; Lane 4: 7284, 2464, 1802).
(TIF)

**S2 Fig. Linearity of linkage and Poisson-multinomial models for simulated plasmid genome integrity.** Fitted values versus residuals plotted for each model.
(EPS)

**S3 Fig. Linearity of linkage and Poisson-multinomial models for simulated rAAV genome integrity.** Fitted values versus residuals plotted for each model.
(EPS)

**S1 Table. Linear fit information for linear models plotted in Fig 3D.**
(PDF)

**S2 Table. Linear fit information for linear models plotted in Fig 6.**
(PDF)

**S1 Raw images. Raw image of S1 Fig.**
(PDF)

## Acknowledgments

We thank Biogen's NGS & Genetic Technologies Lab for assistance with sequencing rAAV material, and Chao-Jung (Julie) Wu for sequence analysis. We thank Romi Admanit and Svetlana Bergelson for analytical review and resource management.

## Author Contributions

**Conceptualization:** Xiaohui Zhao, Yu Wang, Marina Feschenko, Cullen Mason.

**Data curation:** Lauren Tereshko, Tinchi Lin.

**Formal analysis:** Lauren Tereshko, Xiaohui Zhao, Jake Gagnon, Tinchi Lin.

**Funding acquisition:** Marina Feschenko, Cullen Mason.

**Investigation:** Xiaohui Zhao, Trevor Ewald.

**Methodology:** Xiaohui Zhao, Jake Gagnon.

**Project administration:** Lauren Tereshko, Xiaohui Zhao, Marina Feschenko, Cullen Mason.

**Resources:** Jake Gagnon, Tinchi Lin.

**Software:** Tinchi Lin.

**Supervision:** Marina Feschenko, Cullen Mason.

**Validation:** Lauren Tereshko, Jake Gagnon, Tinchi Lin.

**Visualization:** Lauren Tereshko.

**Writing – original draft:** Lauren Tereshko.

**Writing – review & editing:** Lauren Tereshko, Xiaohui Zhao, Jake Gagnon, Tinchi Lin, Cullen Mason.

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
