## [Decision Letter · Decision Letter 0]

9 Aug 2023

PONE-D-23-22796A novel method for quantitation of AAV genome integrity and residual DNAs using duplex digital PCRPLOS ONE

Dear Dr. Tereshko,

Thank you for submitting your manuscript to PLOS ONE. After careful consideration, we feel that it has merit but does not fully meet PLOS ONE’s publication criteria as it currently stands. Therefore, we invite you to submit a revised version of the manuscript that addresses the points raised during the review process.

We look forward to receiving your revised manuscript.

Kind regards,

Ruslan Kalendar

Academic Editor

PLOS ONE

Journal Requirements:

"This work was sponsored by Biogen.

All experimental design, data collection, analyses, and preparation of the manuscript were performed by Biogen employees."

4. We are unable to open your Supporting Information file [File Name]. Please kindly revise as necessary and re-upload.

Reviewers' comments:

Reviewer's Responses to Questions

**Comments to the Author**

1. Is the manuscript technically sound, and do the data support the conclusions?

Reviewer #1: Yes

Reviewer #2: Yes

2. Has the statistical analysis been performed appropriately and rigorously? 

Reviewer #1: Yes

Reviewer #2: N/A

3. Have the authors made all data underlying the findings in their manuscript fully available?

Reviewer #1: Yes

Reviewer #2: Yes

4. Is the manuscript presented in an intelligible fashion and written in standard English?

Reviewer #1: Yes

Reviewer #2: Yes

5. Review Comments to the Author

Reviewer #1: 

Everybody should use this analysis method in the conditions described or be inspired to likewise improve upon the poor mathematics that somehow slipped into usage.

Line 35. A colleague commented to me that long read methods including NGS methods are inferior in precision to dPCR when quantitating amounts of species. Better precision is a technical advantage which might recommend dPCR even if the practical reasons given relating to cost, effort, data analysis and amount of material. My second hand opinion is that even if NGS were improved to be faster, data analysis automated by better software, and miniaturized to use less sample, that primer based methods will still be preferred for higher precision. I suggest that you add a sentence or two based on your experience of NGS methods. First, a statement that NGS supports the method development of dPCR primers. And then a statement of the technical reasons why primer based dPCR can achieve higher precision than NGS methods.

Line 320 Early discussions about electron microscopy use in the field gave me the opinion that Cryo EM worked much better than TEM but the industry tended to use TEM only because of cost despite its technical inferiority. The study in a paper by Werle et al that compared different methods on an unusual AAV sample with very low or missing partially empty capsids showed that even with no partially empty capsids present, TEM has precision issues. (Werle AK, Powers TW, Zobel JF, Wappelhorst CN, Jarrold MF, Lyktey NA, Sloan CDK, Wolf AJ, Adams-Hall S, Baldus P, Runnels HA. Comparison of analytical techniques to quantitate the capsid content of adeno-associated viral vectors. Mol Ther Methods Clin Dev. 2021 Sep 1;23:254-262. doi: 10.1016/j.omtm.2021.08.009. PMID: 34703846; PMCID: PMC8505359.) I suggest adding the Werle reference here and emphasizing that even in the simplest systems, TEM and other orthogonal methods have deficiencies in precision and dynamic range when quantitating percent intact genomes of AAV samples. (For instance, the EM they used always sees some empty or some full capsids even when the tested sample was monodisperse. Image recognition software is a bit imprecise.) The Werle paper also re-enforces the idea that all methods have technical and data analysis limitations that often require orthogonal information to correct. Too often instrument manufacturers distort the science by pressing the opinion that the “customer” will implement only one assay and using pecuniary pseudo-truths try to sell only one instrument.

I commend the authors on using theoretical considerations, on very well characterized plasmid derived samples, and on AAV samples controlling the temperature stress as a parameter. This was an excellent experimental plan. The scheme to present diagrams and figures to visualize a mainly mathematical point, particularly the use of the %recovery parameter, was excellent.

Reviewer #2: 

Recombinant AAV are highly efficient viral vectors for in vivo gene transfer. After decades of research and development, AAV-derived drugs are currently on the market for the treatment of genetic diseases. However, vector production processes do not guarantee homogeneous products. Whereas the holy grail being one AAV capsid containing one whole therapeutic cassette, most of AAV batches contain empty, partial, full and overloaded capsids. Thus, careful control of AAV genome integrity with accurate methods is of utmost importance.

In this manuscript, Lauren Tereshko and collaborators have used multiplex droplet digital PCR to assess rAAV genome integrity. The originality of the manuscript resides in the comparison of different calculation formulas to analyze genome integrity ddPCR data. Interestingly, they have proposed a new and Poisson-multinomial model that improves the accuracy and quantifiable range of duplex ddPCR assays to determine rAAV genome intergrity.

Only minor comments should be addressed:

- Introduction:

(lines 30-34) Add citations for rAAV heterogeneity, residual DNA and purification process statements.

(lines 58-59) Clarify the sentence “We compare the accuracy … genomes of 0-100% across a range of concentrations”. I assume 0-100% in length?

- Material and Methods must be fleshed out.

Please provide primers and probe sequences and PCR cycles conditions.

How was ITR restriction digestion performed?

How genome fragments were generated, verified and quantified? Restriction enzyme sites can be illustrated on Fig1. Add an agarose gel with plasmid fragments as supplemental figure.

Give more details about NGS (library preparation, injected quantity, bioinformatics analysis)

- Results:

Double stranded DNA is not degraded by 95°C heat (doi: 10.1089/dna.2013.2056). Is there another interpretation of your data? Heat effect on AAV single-stranded DNA? Fragments reannealing?

A verification of AAV DNA degradation by heat using an orthogonal method (microfluidic capillary electrophoresis…) may be interesting.

6. PLOS authors have the option to publish the peer review history of their article (what does this mean?). If published, this will include your full peer review and any attached files.

Reviewer #1: **Yes: **David B Hayes

Reviewer #2: **Yes: **Magalie Penaud-Budloo

---

## [Author Response · Author response to Decision Letter 0]

22 Sep 2023

Please note the responses below are included in our “cover letter” and "response to reviewers" documents.

Cover letter:

Dear PLOS ONE Editors,

Thank you for your thoughtful review of our research article titled “A novel method for quantitation of AAV genome integrity using duplex digital PCR”. We appreciate the opportunity to incorporate your suggestions into the manuscript. We believe the changes detailed in our Response to Reviewers have improved the overall clarity and quality of the article.

We acknowledge that there were issues with our Supporting Information files. We have moved the captions for these files to the end of the revised manuscript. We have revised and re-uploaded all figure files. We have also reviewed our References list and verified that no retracted publications are cited. Changes to the reference list are noted in the Response to Reviewers. 

After reviewing the Journal Requirements, we have updated the formatting of our manuscript to align with PLOS ONE’s guidelines. In addition to our Response to Reviewers, we would like to amend the following submission sections:

1. Authorship credits: In our first submission, we accidentally omitted an author. We would like to amend the authorship credits to include Yu Wang for Conceptualization of the project.

2. Role of Funder statement: We would like to amend our statement to, "This work was sponsored by Biogen. The funders had no role in study design, data collection and analysis, decision to publish, or preparation of the manuscript. All experimental design, data collection, analyses, and preparation of the manuscript were performed by Biogen employees."

3. Data Availability statement: We have moved the datasets, R scripts underlying our figures, and the code for our Shiny app to a public GitHub repository. We would like to amend our statement to, “All data files and R code are available for download from GitHub at https://github.com/tlin-biogen/genome-integrity-public”

Response to Reviewers:

We thank the reviewers for their insights and suggestions. We feel that they greatly improved the clarity of the manuscript and added context to how the new methodology relates to orthogonal methods for AAV capsid and genome characterization. Please find our responses to each point interspersed with the reviewers’ comments (italicized) below:

Response to reviewer #1:

Line 35. A colleague commented to me that long read methods including NGS methods are inferior in precision to dPCR when quantitating amounts of species. Better precision is a technical advantage which might recommend dPCR even if the practical reasons given relating to cost, effort, data analysis and amount of material. My second hand opinion is that even if NGS were improved to be faster, data analysis automated by better software, and miniaturized to use less sample, that primer based methods will still be preferred for higher precision. I suggest that you add a sentence or two based on your experience of NGS methods. First, a statement that NGS supports the method development of dPCR primers. And then a statement of the technical reasons why primer based dPCR can achieve higher precision than NGS methods.

We appreciate Reviewer 1’s point about the limitations of the precision of NGS. We clarified our statement on the limitations of NGS in the introduction (lines 44-47). We also provided statements around the benefits of the accuracy/precision/sensitivity of duplex dPCR relative to NGS to the discussion section (lines 366-383).

Line 320 Early discussions about electron microscopy use in the field gave me the opinion that Cryo EM worked much better than TEM but the industry tended to use TEM only because of cost despite its technical inferiority. The study in a paper by Werle et al that compared different methods on an unusual AAV sample with very low or missing partially empty capsids showed that even with no partially empty capsids present, TEM has precision issues. (Werle AK, Powers TW, Zobel JF, Wappelhorst CN, Jarrold MF, Lyktey NA, Sloan CDK, Wolf AJ, Adams-Hall S, Baldus P, Runnels HA. Comparison of analytical techniques to quantitate the capsid content of adeno-associated viral vectors. Mol Ther Methods Clin Dev. 2021 Sep 1;23:254-262. doi: 10.1016/j.omtm.2021.08.009. PMID: 34703846; PMCID: PMC8505359.) I suggest adding the Werle reference here and emphasizing that even in the simplest systems, TEM and other orthogonal methods have deficiencies in precision and dynamic range when quantitating percent intact genomes of AAV samples. (For instance, the EM they used always sees some empty or some full capsids even when the tested sample was monodisperse. Image recognition software is a bit imprecise.) The Werle paper also re-enforces the idea that all methods have technical and data analysis limitations that often require orthogonal information to correct. Too often instrument manufacturers distort the science by pressing the opinion that the “customer” will implement only one assay and using pecuniary pseudo-truths try to sell only one instrument.

We agree with the reviewer’s suggestion to further emphasize the distinction between capsid characterization methods and genome characterization methods. We added a statement to the introduction to highlight the gap in capsid methodologies for characterizing genomes (lines 38-47). We have also added to the discussion as suggested, to now emphasize the value and necessity of employing orthogonal methods for characterizing capsid content and added several references, including the suggested reference Werle et al 2021 (lines 366-383).

I commend the authors on using theoretical considerations, on very well characterized plasmid derived samples, and on AAV samples controlling the temperature stress as a parameter. This was an excellent experimental plan. The scheme to present diagrams and figures to visualize a mainly mathematical point, particularly the use of the %recovery parameter, was excellent.

We thank the reviewer for these kind words regarding the experimental plan and presentation of the data. 

Response to Reviewer #2:

In this manuscript, Lauren Tereshko and collaborators have used multiplex droplet digital PCR to assess rAAV genome integrity. The originality of the manuscript resides in the comparison of different calculation formulas to analyze genome integrity ddPCR data. Interestingly, they have proposed a new and Poisson-multinomial model that improves the accuracy and quantifiable range of duplex ddPCR assays to determine rAAV genome integrity.

Only minor comments should be addressed:

(lines 30-34) Add citations for rAAV heterogeneity, residual DNA and purification process statements.

We added several citations for rAAV heterogeneity and process impurities to the introduction (lines 36-38).

(lines 58-59) Clarify the sentence “We compare the accuracy … genomes of 0-100% across a range of concentrations”. I assume 0-100% in length?

We removed the 0-100% statement as it added confusion. The range of 0-100% simulated integrities corresponds to ranges in the percentage of the presence of both genetic termini from the population of template DNA molecules present. We added details regarding the preparation of mock integrity samples throughout the manuscript (see in particular lines 85-94). We hope these edits clarify that duplex dPCR reactions serve as a proxy for genome integrity by measuring the binary presence or absence of each genetic terminus and therefore do not directly measure genome length. 

Material and Methods must be fleshed out. Please provide primers and probe sequences and PCR cycles conditions. How was ITR restriction digestion performed? Give more details about NGS (library preparation, injected quantity, bioinformatics analysis)

We have added significantly to the Materials and Methods section and now provide details for: DNA material identity, primer/probe sequences, thermocycling program, ITR digestion, NGS library preparation, sequencing, and analytical pipeline. We have added references to this section as needed.

How genome fragments were generated, verified and quantified? Restriction enzyme sites can be illustrated on Fig1. Add an agarose gel with plasmid fragments as supplemental figure.

Figure 1 was updated for clarity and includes the restriction sites. We have added a supplemental figure (S1 Fig) confirming the digestion of pAAV with MfeI alone and MfeI/NheI together. The image of the gel shows that the digested plasmid has the expected banding pattern for “intact” and “fragmented” genomes. 

Double stranded DNA is not degraded by 95°C heat (doi: 10.1089/dna.2013.2056). Is there another interpretation of your data? Heat effect on AAV single-stranded DNA? Fragments reannealing? A verification of AAV DNA degradation by heat using an orthogonal method (microfluidic capillary electrophoresis…) may be interesting.

We appreciate Reviewer 2’s suggestion to verify the effects of heat on AAV DNA. In contrast to Karni et al 2013 (doi: 10.1089/dna.2013.2056), which tested the effects of heat on plasmid DNA, we have used linear dsDNA and ssDNA which have been shown to be less stable than circular DNA (Marguet and Forterre 1994 (https://doi.org/10.1093/nar/22.9.1681), Lindahl and Nyberg 1972 (https://doi.org/10.1021/bi00769a018)). Although Karni et al 2013 did not see significant degradation of plasmid DNA at 95°C, this was tested only with dry DNA for 5 minutes. In their discussion section, the authors state “In an aqueous solution, it is not possible to determine the degradation temperature using our method, since applying pressure on the DNA solution causes the DNA to be more sensitive to heat, and therefore the DNA degrades already above 90°C.” In addition to the two mentioned above, multiple other peer-reviewed publications have previously reported thermal degradation of dsDNA and ssDNA in aqueous solution. Our reference section has been updated to expand on these citations (see references 22, 25-31).

To address Reviewer 2’s suggestion, we have performed an alternative experiment. Unfortunately, we do not have enough of the ssDNA AAV material that was used in Figure 7 to visualize the degradation by electrophoresis. We instead linearized AAV vector plasmid DNA by restriction digest with BamHI, performed heat stress, and resolved the samples by agarose gel electrophoresis. Below are the results of incubating 500 ng samples of linearized plasmid DNA for 0, 5, 15 or 30 minutes at 95°C. The control lane (0 minutes at 95°C) shows the majority of DNA is of the expected length (~12 KB). Upon heat treatment, the band gradually becomes smaller. At 15 and 30 minutes at 95°C, smearing below the band can be seen at increasingly small sizes.

 While the data here demonstrate the thermal instability of linearized plasmid DNA, this does not directly correlate to the manuscript as Fig 7 uses AAV material; we therefore did not include this as a supplemental figure. However, we believe these data combined with our results from Fig 7, and the additional cited publications support our conclusion that rAAV DNA degrades into smaller fragments upon heat treatment thereby reducing the number of intact templates. Although re-annealing of ssDNA fragments could potentially occur, this alone would not affect measured genome integrities, as it would not change the ratio of the number of 5’ and 3’ targets that are present during PCR amplification cycles. Furthermore, we do not see changes in the concentration of ddPCR targets in the raw data across incubation times, which would be expected to decrease if annealing occurred.

---

## [Decision Letter · Decision Letter 1]

10 Oct 2023

A novel method for quantitation of AAV genome integrity using duplex digital PCR

PONE-D-23-22796R1

Dear Dr. Tereshko,

We’re pleased to inform you that your manuscript has been judged scientifically suitable for publication and will be formally accepted for publication once it meets all outstanding technical requirements.

Kind regards,

Simone Agostini, Ph.D.

Academic Editor

PLOS ONE

Reviewers' comments:

Reviewer's Responses to Questions

**Comments to the Author**

1. If the authors have adequately addressed your comments raised in a previous round of review and you feel that this manuscript is now acceptable for publication, you may indicate that here to bypass the “Comments to the Author” section, enter your conflict of interest statement in the “Confidential to Editor” section, and submit your "Accept" recommendation.

Reviewer #1: All comments have been addressed

Reviewer #2: All comments have been addressed

2. Is the manuscript technically sound, and do the data support the conclusions?

Reviewer #1: Yes

Reviewer #2: Yes

3. Has the statistical analysis been performed appropriately and rigorously? 

Reviewer #1: Yes

Reviewer #2: Yes

4. Have the authors made all data underlying the findings in their manuscript fully available?

Reviewer #1: Yes

Reviewer #2: Yes

5. Is the manuscript presented in an intelligible fashion and written in standard English?

Reviewer #1: Yes

Reviewer #2: Yes

6. Review Comments to the Author

Reviewer #1: The comments about the original manuscript asked for some more information and clarification. The authors did a good job adding the requested information and looking up extra references. I support adding some information as supplemental. Also, it seems to me that the initial temperature degradation studies were sufficient for the purpose of the paper. I believe that the extra experiments performed were helpful in supporting the initial results and that the authors also properly discussed why heat degradation studies are perhaps more complicated than one would expect.

Reviewer #2: (line 38) "Current purification processes can efficiently separate empty from full capsids". Please moderate "efficiently". Empty capsids (and even more partial capsids) are hard to completely remove.

7. PLOS authors have the option to publish the peer review history of their article (what does this mean?). If published, this will include your full peer review and any attached files.

Reviewer #1: **Yes: **David B Hayes

Reviewer #2: **Yes: **Dr PENAUD-BUDLOO Magalie

---

## [Editor Report · Acceptance letter]

5 Dec 2023

PONE-D-23-22796R1 

A novel method for quantitation of AAV genome integrity using duplex digital PCR 

Dear Dr. Tereshko:

I'm pleased to inform you that your manuscript has been deemed suitable for publication in PLOS ONE. Congratulations! Your manuscript is now with our production department. 

Kind regards, 

on behalf of

Dr. Simone Agostini 

Academic Editor

PLOS ONE